# Rapid and High-Throughput Determination of Sixteen *β*-agonists in Livestock Meat Using One-Step Solid-Phase Extraction Coupled with UHPLC-MS/MS

**DOI:** 10.3390/foods12010076

**Published:** 2022-12-23

**Authors:** Yonghong Yan, Jun Ning, Xin Cheng, Qingqin Lv, Shuang Teng, Wei Wang

**Affiliations:** 1College of Food Science and Technology, Nanjing Agricultural University, Nanjing 210095, China; 2National Center of Meat Quality and Safety Control, Nanjing Agricultural University, Nanjing 210095, China; 3Key Laboratory of Animal Products Processing, Ministry of Agriculture and Rural Affairs, Nanjing 210095, China; 4Jiangsu Collaborative Innovation Center of Meat Production and Processing, Quality and Safety Control, Nanjing Agricultural University, Nanjing 210095, China; 5WENS Foodstuff Group Co., Ltd., Yunfu 527400, China; 6The Center for Agri-Food Quality & Safety, Ministry of Agriculture and Rural Affairs, Beijing 100125, China

**Keywords:** *β*-agonist, one-step solid-phase extraction, UHPLC-MS/MS, livestock meat

## Abstract

*β*-agonists are illegally added to animal feed because they can greatly increase carcasses’ leanness, which impairs the safety of animal-derived foods and indirectly endangers human health. This study aimed to develop an ultrahigh-performance liquid chromatography-tandem mass spectrometry (UHPLC-MS/MS) method for determining sixteen *β*-agonists in livestock meat. The homogenized samples were subjected to enzymatic hydrolysis using *β*-glucuronidase/sulfatase at 40 °C for 2 h, extracted with acetonitrile containing 1% acetic acid (*v*/*v*), and purified by the one-step Qvet-AG extraction column. The residue was redissolved by 0.1% aqueous formic acid/methanol (9:1, *v*/*v*) after blow-drying by nitrogen, and then determined by UHPLC-MS/MS. The results demonstrated that the well linearity was in the range of 0.1–50 μg/L with the correlation coefficient (R^2^) ≥0.9928, and the limits of detection (LOD) and quantification (LOQ) were 0.01–0.11 μg/kg and 0.04–0.38 μg/kg, respectively. With intraday and interday relative standard deviations (RSDs) being less than 10%, the average recoveries of pork, beef, and lamb at various spiked levels ranged from 62.62–115.93%, 61.35–106.34%, and 62.00–111.83%, respectively. In conclusion, the established method is simple, efficient, sensitive, and suitable for the simultaneous detection of several *β*-agonist residues in livestock meat.

## 1. Introduction

Livestock meat has traditionally been a fundamental part of the human diet because of its tremendous nutritional value. With the improvement of living standards, meat consumption shows a trend of annual growth. In recent years, people’s views on health have also changed, moving away from high-fat and high-energy diets toward low-fat and low-sugar foods. This has an impact on traditional livestock farming, such as lean pigs being more favored by consumers. *β*-agonists are a class of phenylethanolamine compounds with an adrenergic function; they were initially developed to treat uterine, respiratory, and cardiovascular diseases in humans and animals [1]. Later, studies found that *β*-agonists significantly improved lean muscle mass of carcasses and the feed conversion ratio of animals when taken at doses 5 to 10 times higher than therapeutic doses [2]. Because of their huge feeding benefits, the illicit abuse of *β*-agonists in animal feed occurs often, especially in livestock animals such as pigs, cattle, and sheep, which are more difficult to regulate. However, the chemical structures of *β*-agonists are stable and frequently difficult to be degraded in animals, making them easily residual in the human body through the food chain. Serious adverse reactions, such as muscle tremors, dizziness, rapid heartbeat, acute poisoning, and even direct death, could occur when the body accumulates a certain amount [3,4,5]. Some organizations and nations have created pertinent laws regarding the usage of *β*-agonists in order to safeguard the health of consumers. For example, the European Union (EU, 1996) banned all *β*-agonists used for growth-promoting effects in animals and established maximum residue limits (MRLs) in various animal tissues at the same time [6]. Additionally, China also explicitly prohibited the use of *β*-agonists in animal feed and drinking water, and made them target substances for the supervision and routine monitoring of the nation’s agricultural quality and safety [7]. Some nations, such as the United States (US), Canada, and Japan, limited residue standards, but did not outright forbid the use of all *β*-agonists [8]. It is clear that the development of a scientifically accurate method for detecting *β*-agonists would help to ensure the safety of animal-derived food.

Immunological and instrumental methods are currently the primary analytical methods for *β*-agonists in animal-derived foods. Enzyme-linked immunosorbent assays (ELISA) [9] and colloidal gold immunochromatography assays (GICA) [10] are examples of immunological techniques. Instrumental methods include capillary electrophoresis (CE) [11], gas chromatography-tandem mass spectrometry (GC-MS/MS) [12], liquid chromatography-tandem mass spectrometry (LC-MS/MS) [13]. In addition, Raman spectroscopy [14], the sensor method [15], and the biochip method [16] have been used recently. Due to the challenges in preparing antigens or antibodies, their limited sensitivity, and their propensity for producing false positive results, immunological methods are frequently utilized only for routine screening detections in the market [17]. New detection techniques such as sensors are still less applied and need to be further studied. Ultrahigh-performance liquid chromatography-tandem mass spectrometry (UHPLC-MS/MS), which belongs to the LC-MS/MS method, has become the mainstream qualitative and quantitative approach for the determination of *β*-agonists due to its shorter analysis time, higher sensitivity, and better accuracy [18,19,20]. However, the existing pretreatment process of the UHPLC-MS/MS method has the disadvantages of tedious and time-consuming operation, high reagent consumption, and strong matrix interference with poor recovery. In this study, by optimizing the pretreatment and instrument conditions, a rapid and high-throughput determination of sixteen *β*-agonists in livestock meat was established by using one-step solid-phase extraction coupled with UHPLC-MS/MS, which not only is of great significance for safeguarding animal food safety, and but also could provide a reference for production monitoring, daily batch detection, and the confirmatory analysis of *β*-agonists in livestock meat.

## 2. Materials and Methods

### 2.1. Chemicals and Reagents

Sixteen *β*-agonist standards and five internal standards, including clenbuterol (CLB), mabuterol (MAB), brombuterol (BRM), clenpenterol (CLP), ractopamine (RAC), salbutamol (SAL), terbutaline (TEB), formoterol (FOM), isoxsuprine (ISS), ritodrine (RIT), phenylethanolamine A (PEA), clorprenaline (CLO), tulobuterol (TUL), bambuterol (BAM), cimbuterol (CIB), hydroxymethyl clenbuterol (H-CLB), clenbuterol-D9 (CLB-D9), ractopamine-D6 (RAC-D6), salbutamol-D3 (SAL-D3), phenylethanolamine A-D3 (PEA-D3), and bambuterol-D9 (BAM-D9), were obtained from Alta Scientific Co., Ltd. (Tianjin, China). All standards were 100 mg/L in concentration and the purity was ≥99%. *β*-glucuronidase/sulfatase (>100,000 units/mL) was purchased from Anpel Laboratory Technologies Inc. (Shanghai, China). High-performance liquid chromatography-grade methanol and acetonitrile were supplied by Merck (Darmstadt, Germany). Formic acid (HPLC-grade), acetic acid (analytical grade), ammonium acetate, and sodium chloride (NaCl) were purchased from Sinopharm Group Chemical Reagent Co., Ltd. (Shanghai, China). Ultrapure water was supplied by the Sartorius Arium^®^ pro system (Sartorius, Göttingen, Germany).

### 2.2. Sample Collection 

Blank samples used in the validation process were provided by the Supervision, Inspection, and Testing Center for Quality of Meat Products (Nanjing, China). A total of 30 commercially available livestock meat products (sample numbers: P-1, P-2, P-3, P-4, P-5, P-6, P-7, P-8, P-9, P-10, B-1, B-2, B-3, B-4, B-5, B-6, B-7, B-8, B-9, B-10, L-1, L-2, L-3, L-4, L-5, L-6, L-7, L-8, L-9, L-10) tested in the actual sample were obtained from three provinces in China. The letters P, B, and L represent pork, beef, and lamb, respectively. Numbers 1–4 represent samples taken from Jiangsu, the 5–7 from Heilongjiang, and 8–10 from Guangxi. 

### 2.3. Standard Solution Preparation

A total of 100 mg/L of sixteen standards and five internal standards as standard stock solutions was stored at −20 °C for up to six months. Mixed standard solution at the concentration of 1.0 mg/L was prepared by diluting the sixteen standard stock solutions with methanol. Mixed internal standard solution at the concentration of 1.0 mg/L was prepared by diluting the five internal standard stock solutions with methanol as well. The aforementioned mixed standard solutions were stored at −20 °C for one month. According to experimental needs, the mixed standard solution and the mixed internal solution were diluted by methanol to obtain 100 μg/L of the mixed standard working solution and mixed internal standard working solution, respectively, and stored at 4 °C until use.

### 2.4. UHPLC-MS/MS Instrumentation and Operating Conditions

Thermo Scientific Vanquish ultrahigh-performance liquid chromatography coupled to Thermo Scientific TSQ Quantis mass spectrometer (Thermo Fisher Scientific, Waltham, MA, USA) was used. Separation was performed with the Thermo Hypersil GOLD aQ column (100 mm × 2.1 mm, 1.9 μm; Thermo Fisher Scientific, Waltham, MA, USA). The column temperature was 30 °C, and the sample injection volume was 3 μL. Mobile phase A was 0.1% aqueous formic acid (*v*/*v*), mobile phase B was methanol containing 0.1% formic acid (*v*/*v*), and the flow rate was held at 0.3 mL/min throughout the analysis. Gradient conditions were 0–2.0 min (95% A), 2.1–5.0 min (70% A), 5.1–9.0 min (60% A), 9.1–12.0 min (10% A), 12.1–13.0 min (95% A), and held at this level for 3 min so that the system could re-equilibrate before the next injection.

The MS/MS was equipped with an ESI source operating in the positive ionization mode [M+H]^+^ and the data were collected in multiple reaction monitoring (MRM) mode. The optimized electrospray ionization parameters were as follows: capillary voltage, 3.5 kV; ion source temperature, 450 °C; ion transfer tube temperature, 300 °C; vaporizer temperature, 280 °C; sheath gas, 38 Arb; auxiliary gas, 4 Arb.

### 2.5. Sample Preparation

Pork, beef, and lamb samples (2.00 ± 0.02 g) were accurately weighed into a 50 mL polypropylene centrifuge tube after being chopped and homogenized using an HM6300 intelligent homogenizer (Lab Precision Beijing Technology Co., Ltd., Beijing, China). Samples were spiked with 50 μL of the mixed internal standard working solution (100 μg/L). A total of 10 mL of 0.2 mol/L ammonium acetate buffer solution (pH 5.2) and 40 μL *β*-glucuronidase/sulfatase was added and mixed (Vortex Genius 3, IKA-Werke GmbH & Co., Staufen, Germany). It was incubated for 2 h at 40 °C (TW20, Julabo GmbH, Seelbach, Germany). After being cooled to room temperature, the homogenate was then centrifuged at 10,621× *g* for 5 min (Centrisart^®^ D-16C, Sartorius, Göttingen, Germany), and the supernatant was transferred to another 50 mL polypropylene centrifuge tube. Then, 10 mL of acetonitrile containing 1% acetic acid (*v*/*v*) and 2 g of NaCl was successively added into the supernatant. Next, the mixture was fully vortexed for 5 min and centrifuged at 10,621× *g* for 5 min again, taking 2 mL of clear supernatant passed through the Qvet-AG one-step solid-phase extraction column (6 mL–2 g, Shimadzu, Kyoto, Japan) at a rate of about 1 drop/second for further purification. The entire filtrate was then collected directly in a test tube, which then was evaporated to dryness in a 40 °C water bath under a gentle flow of nitrogen (N-EV AP-11 nitrogen evaporator, Organomation, Berlin, MA, USA), and redissolved to 1 mL with 0.1% aqueous formic acid/methanol (9:1, *v*/*v*). After mixing thoroughly, the solution was filtered through the 0.22 μm hydrophilic filter membrane (Agilent Technologies, Santa Clara, CA, USA) for UHPLC-MS/MS analysis. The schematic diagram of the main experimental operation is shown in Figure 1.

### 2.6. Method Validation

#### 2.6.1. Matrix Effect Evaluation

The Matrix Effect (ME), which has an impact on the accuracy of the quantitative results, is the interference of other components in the sample with the analytical process of the target compound [21]. According to the established sample processing method, the matrix-matched standard curves (the seven concentration levels of compounds were 0.1, 0.5, 1.0, 2.0, 5.0, 10.0, and 50.0 μg/L) were obtained by the blank samples and compared with the reagent standard curves obtained by redissolved solution (0.1% aqueous formic acid/methanol, 9:1, *v*/*v*). The MEs were evaluated by calculating the relationship between the slope ratio of the matrix-matched standard curve and the reagent standard curve, and the formula is: (1)$$\mathrm{ME}\;\left(\%\right)=\frac{K_{a}}{K_{b}}\times 100$$
where *k_a_* is the slope of the matrix-matched standard curve; *k_b_* is the slope of the reagent standard curve. When ME > 1, it indicates a matrix enhancement effect; when ME < 1, it indicates a matrix inhibition effect, and when ME = 1, then there is no matrix interference [22].

#### 2.6.2. Determination of Linearity, Limit of Detection, and Limit of Quantification

Based on the established UHPLC-MS/MS method, the prepared matrix-matched standard working solutions were determined on the machine under optimized instrumental conditions. The standard curve was plotted by linear fitting with the ratio of compound concentration (μg/L) to the concentration of the internal standard (μg/L) as the X-axis, and the ratio of the peak area of quantitative ion chromatography for each compound to the peak area of the corresponding internal standard as the Y-axis. From this, the regression equations were obtained and the corresponding squared correlation coefficients (R^2^) were calculated.

The limits of detection (LODs) and limits of quantification (LOQs) for the target compounds were assessed by testing a series of spiked blank samples. The spiked concentration at signal-to-noise (S/N) ≥ 3 was used as LOD of the method, while LOQ of the method was the spiked concentration at S/N ≥ 10 [23].

#### 2.6.3. Recovery and Precision Test

The accuracy (expressed as recovery) and the precision (as relative standard deviation, RSD) of the analytical method were evaluated using spiked blank samples at three concentration levels (0.5, 1.0, and 5.0 μg/kg), and each concentration was set in six parallels. The analysis was repeated three times (three different days), and the average recovery was calculated along with intraday and interday RSD. The results were used to assess the accuracy, reproducibility, and stability of the method. The recovery and RSD were calculated as follows:(2)$$\mathrm{recovery}\;\left(\%\right)=\frac{2.5\times C_{E}}{C_{S}}\times 100$$
(3)$$\mathrm{RSD}\;\left(\%\right)=\frac{\mathrm{SD}}{C_{A}}\times 100$$
where 2.5 is the conversion multiplier, *C_E_* (μg/L) is the experimental concentration determined from the calibration curve, *C_S_* (μg/kg) is the spiked concentration, SD (μg/L) is the standard deviation, and *C_A_* (μg/L) is the average of experimental concentration determined from the calibration curve.

### 2.7. Analysis of Actual Sample

Thirty samples of pork, beef, and lamb were randomly chosen from three different Chinese provinces and tested for sixteen *β*-agonists using the method determined in this work. To further confirm the feasibility and accuracy of the method, sixteen compounds were quality controlled by blank spiking during the assay. 

### 2.8. Data Analysis

TraceFinder 4.1 software (Thermo Fisher Scientific, Waltham, MA, USA) was used for data acquisition and processing, and OriginPro software (2022b, OriginLab Inc., Northampton, MA, USA) was used for plotting. Three parallels were performed for each experiment, and data were expressed as mean ± standard deviation.

## 3. Results and Discussion

### 3.1. Optimization of UHPLC-MS/MS Conditions

#### 3.1.1. Optimization of Chromatographic Condition

The majority of the research currently in publication uses reversed-phase columns for the separation of *β*-agonists, which are moderately or strongly polar compounds [24,25]. In this test, we compared the separation performances of three columns: the Acquity UPLC HSS C18 column (150 mm × 2.1 mm, 1.7 μm; Waters, Milford, MA, USA), the Accucore^TM^ aQ column (150 mm × 2.1 mm, 2.6 μm; Thermo Fisher Scientific, Waltham, MA, USA), and the Hypersil GOLD aQ column (100 mm × 2.1 mm, 1.9 μm; Thermo Fisher Scientific, Waltham, MA, USA). The Hypersil GOLD column was chosen due to its outstanding aqueous mobile phase stability, strong retention capabilities, and excellent separation properties for polar compounds.

To better separate the sixteen *β*-agonists, different mobile phases such as water-methanol, water-acetonitrile, 0.1% aqueous formic acid (*v*/*v*)-methanol containing 0.1% formic acid (*v*/*v*), and 0.1% aqueous formic acid (*v*/*v*)-acetonitrile containing 0.1% formic acid (*v*/*v*) were tried. The results demonstrated that adding a certain acid to the mobile phase could enhance the ionization efficiency and make the target compounds easier to adsorb and elute from the column. Moreover, since the elution capacity of methanol is weaker than that of acetonitrile, *β*-agonists with similar polarity enable elution in segments. Sharp symmetrical peak patterns can be obtained with a high signal responsiveness due to the solvent’s consistency. Consequently, 0.1% aqueous formic acid (*v*/*v*)-methanol containing 0.1% formic acid (*v*/*v*) was used as the mobile phase composition.

It is challenging to separate these sixteen substances using isocratic elution since the compounds have similar polarities. In order to obtain the best possible separation, the gradient elution procedure was finally determined by adjusting the elution intensity of the mobile phase. The sixteen target compounds were well separated under the determined chromatographic condition, and the analysis time was 16 min, which is faster than the method in the Chinese standard GB/T 22286-2008 for the detection of eleven *β*-agonists in 26 min. The ultrahigh-performance liquid chromatogram of the mixed standard solution is shown in Figure 2.

#### 3.1.2. Optimization of Mass Spectrometry Condition

Sixteen single standard solutions and five individual internal standard solutions of 100 μg/L were injected into the mass spectrometer at a flow rate of 3 μL/min by using a needle pump. The Q1 full scan was turned on in positive ion mode to determine the molecular ion of each standard. Then, the instrument scanned in SIM Q3; for each target compound, the two pairs of characteristic ion pairs with the highest response were selected as qualitative and quantitative ion pairs. Finally, various mass spectral parameters were optimized in the MRM mode (Table 1).

### 3.2. Optimization of Pretreatment Conditions

#### 3.2.1. Optimization of Enzymatic Condition

Enzymolysis and acid hydrolysis are typically utilized to achieve the accurate quantification of *β*-agonists in organisms [26]. Glucuronidase is also mostly used for enzymolysis; however, the enzymatic effects are greatly influenced by the temperature and time of the enzymolysis. According to the reported literature [1,27,28], the blank spiking recovery was employed as an indicator to compare the effect of different temperature and time on the enzymatic effects. As shown in Table 2, the enzymatic effect at 40 °C for 2 h was comparable to the effect at 37 °C for 12 h. The recovery of seven compounds, including ISS, TUL, RAC, etc., was poor when enzymolysis was performed at 55 °C, suggesting that the temperature affected the enzyme activity and consequently the recovery of the target compounds. As a result, the present method identified 2 h of enzymolysis at 40 °C. Compared with testing methods in the Chinese Standards (GB/T 21313-2007 [19], and SN/T 1924-2011 [20]) using enzymolysis at 37 °C for 12 h, this method effectively reduced the pretreatment time and improved the detection efficiency.

#### 3.2.2. Optimization of Extraction Solvent

*β*-agonists have moderate or strong polarity, and methanol and acetonitrile are typically used as extraction solvents in the residue analysis of such substances, which can reduce the interference of high-fat and high-protein matrices [29]. Based on the spiked recovery of the target compounds, the effects of no-extraction, methanol extraction, and acetonitrile containing 1% acetic acid (*v*/*v*) extraction were compared. Table 3 shows the poor recovery of multiple compounds in the absence of no extraction and methanol extraction. Compared with methanol, acetonitrile brought in lesser polar interfering substances during extraction and provided a better subsequent purification effect. Additionally, the addition of a tiny quantity of acid to acetonitrile was more favorable for the extraction of weakly basic *β*-agonists, and these results are consistent with the published literature [30,31], so the experiment chose acetonitrile containing 1% acetic acid (*v*/*v*) as the extraction solvent.

#### 3.2.3. Optimization of Solid-Phase Extraction Column 

There are still various impurities in the extract to interfere with the targets’ response on the instrument even after extraction with acetonitrile containing 1% acetic acid (*v*/*v*). Further purification is necessary, and solid-phase extraction (SPE) has been widely used as a mature sample pretreatment technique, which can effectively separate the target and impurities. Its advantages over conventional liquid-liquid extraction include lesser solvent consumption, convenient operation, and superior stability [24]. By reviewing the literature, the Oasis MCX column [32,33] (60 mg/3 mL, Waters, Milford, MA, USA), the Oasis HLB column [34,35] (200 mg/3 mL, Waters, Milford, MA, USA), and the Bond Elut C18 column [36,37] (200 mg/3 mL, Agilent Technologies, Santa Clara, CA, USA) were more commonly used in previous *β*-agonists assays. Qvet-AG, as a new one-step veterinary drug pretreatment column, is less utilized at present. The purification effects of the four SPE columns mentioned above for *β*-agonists in this experiment were compared based on the spiked recoveries of the sixteen compounds. As shown in Table 4, the sample had the best purification effect on the Qvet-AG column, and the spiking recoveries of all the compounds met the requirements. Moreover, the Qvet-AG column was chosen for the experiment because, compared with conventional SPE columns, it did not need activation, equilibration, elution, and other steps; the one-step operation efficiently shortened the sample purification time by about 80% at the same time as reducing the amount of reagent consumed.

#### 3.2.4. Optimization of Redissolved Solution

The choice of redissolved solution before sample loading directly affects the sensitivity of the method. The 10 μg/L mixed standard working solution was blown dry under nitrogen and redissolved using different volume ratios of 0.1% aqueous formic acid/methanol (90:10, 70:30, 50:50). It was discovered that the increase in the proportion of the organic phase made the target more soluble and significantly improved the response of the sixteen compounds. However, as illustrated in Figure 3a,b, the precision of the quantitative results is easily impacted by the solvent effects when the ratio of organic phase in the redissolved solution is too high. Therefore, 0.1% aqueous formic acid/methanol (90:10, *v*/*v*) was finally selected as the redissolved solution.

### 3.3. Validation of Analytical Methods

#### 3.3.1. Matrix Effect Evaluation and Elimination

Figure 4 displays the matrix effects of the sixteen compounds in three livestock substrates. It can be seen that most compounds were inhibited by the three matrices, the inhibition rate was different in size, and the matrix effects of different matrices for the same compound were not all the same. Due to the complexity of food matrices and the requirement for trace analysis of the target, this method used the matrix-matched standard curve method and the isotope internal standard method for quantification, which can reduce the error introduced by the pretreatment to improve the accuracy of quantitative analysis results.

#### 3.3.2. Linearity of the Standards Curves, LODs and LOQs

The prepared blank matrix-matched standard solutions were determined online using the established UHPLC-MS/MS method. The results showed a good linearity in the concentration range of 0.1–50 μg/L for the sixteen *β*-agonists with R^2^ ≥ 0.9928, and the linear equations are shown in Table 5. The method LODs ranged from 0.01 to 0.11 μg/kg and the LOQs ranged from 0.04 to 0.38 μg/kg according to the S/N, which was lower than the LODs of *β*-agonists published in the literature [38,39]. These results further indicated that the present method possessed wide linearity, low LOD, and good precision, which was suitable for the determination of *β*-agonists in real animal-derived foods.

#### 3.3.3. Recovery and Precision

The mean recovery ranges for pork, beef, and lamb at different spiked levels were 62.62–115.93%, 61.35–106.34%, and 62.00–111.83%, respectively, and the interday and intraday precision (RSD) of all compounds in the three matrices was 0.66–9.99% and 2.12%–9.98, respectively (Table A1, Table A2, and Table A3). Table A1, Table A2 and Table A3 specifically display the mean recoveries and intraday and interday precision of the sixteen *β*-agonists in the three matrices, which satisfy the requirements of a recovery range of 60–120% and precision of less than 20% [40]. These results indicate that the method has good accuracy and stability for the detection of the sixteen *β*-agonists in livestock meat.

### 3.4. Analyses of Actual Sample

During the determination of 30 commercially available samples, a beef sample (sample number: B-4) revealed the presence of CLB, with a residue of 2.45 ± 0.30 μg/kg and detection rate of 3.33%. The chromatogram and mass spectrogram of the positive clenbuterol sample are shown in Figure 5a,b. The blank samples were used as quality control (QC) samples by spiking during the assay. The CLB was added with 5.0 μg/kg into the QC sample, and the determination value was 4.73 ± 0.08 μg/kg with a good recovery. The identification of the positive sample and the validation results of the QC sample further proved the feasibility and accuracy of the method.

## 4. Conclusions

*β*-agonist residues in livestock meat have a negative impact on human health. With the strengthening of the monitoring of veterinary drug residues worldwide, the requirements for the detection of *β*-agonist residues in animal-derived food have gradually increased as well, and simple, rapid, high-throughput and accurate detection methods have gradually become the development direction. In this study, a UHPLC-MS/MS method was developed for the determination of sixteen *β*-agonists in livestock meat. Optimized enzymatic conditions during sample pretreatment greatly reduced the time of the assay. Following extraction, the sample was purified using the Qvet-AG one-step column, which improved the detection efficiency while also reducing the consumption of reagents. Finally, the sample was used for UHPLC-MS/MS determination, the matrix-matched standard curve method and isotope internal standard method for quantification. The results showed a good linearity in the range of 0.1–50 μg/L for the sixteen *β*-agonists, with R^2^ ≥ 0.9928. The LODs of the method were 0.01–0.11 μg/kg and the LOQs were 0.04–0.38 μg/kg. In the pork, beef, and lamb matrices, the spiking recoveries of the three levels of the *β*-agonists were 62.62–115.93%, 61.35–106.34%, and 62.00–111.83%, respectively. Both the intraday and interday precision fell under 10%. The established method is sensitive and reproducible, with a short analysis time, and high detection quantity, which is suitable for the routine detection and confirmatory analysis of *β*-agonists in animal-derived foods and helps in the regulation of *β*-agonist residues in the market.

## Figures and Tables

**Figure 1 foods-12-00076-f001:**
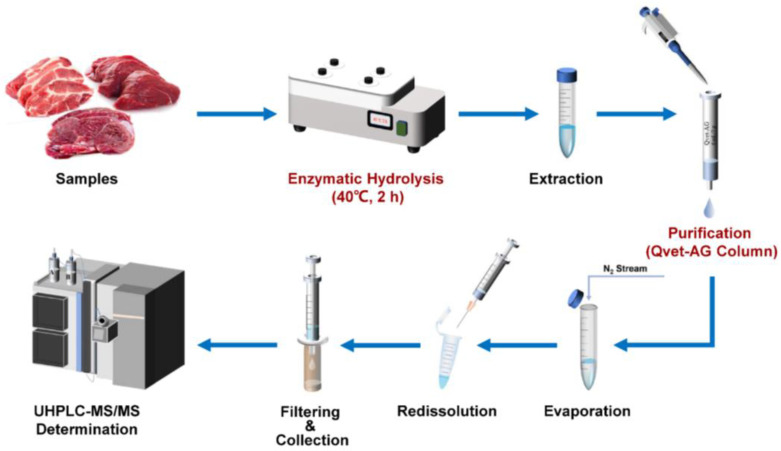
The schematic diagram of main experimental operation for the detection of *β*-agonists by UHPLC-MS/MS.

**Figure 2 foods-12-00076-f002:**
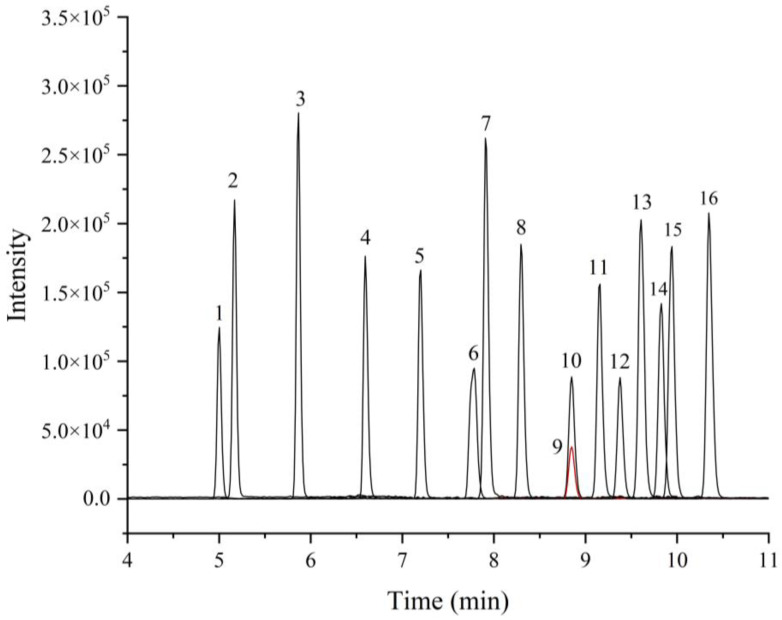
Ultrahigh-performance liquid chromatogram of the mixed standard solution (10 μg/L). Peaks: 1, TEB; 2, SAL; 3, CIB; 4, RIT; 5, H-CLB; 6, RAC; 7, CLO; 8, CLB; 9, PEA; 10, FOM; 11, TUL; 12, BRM; 13, MAB; 14, CLP; 15, ISS; 16, BAM. Although the two substances of PEA and FOM have the same peak time on chromatography, they can be further separated by mass spectrometry.

**Figure 3 foods-12-00076-f003:**
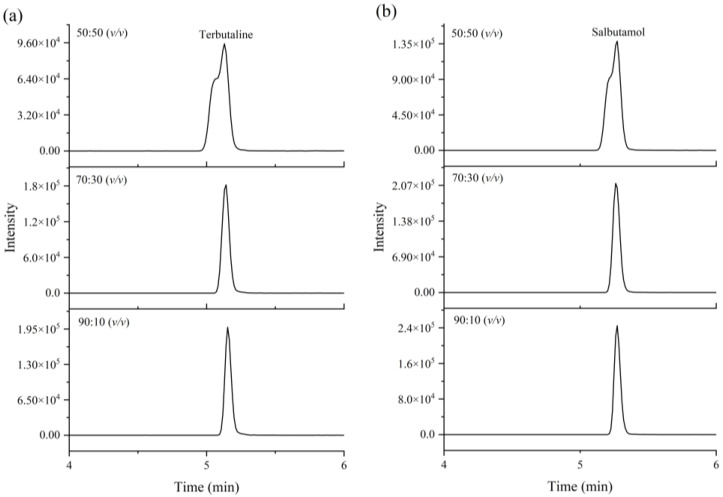
The solvent effect caused by methanol in redissolved solution. (**a**) Comparison of the peak shapes of Terbutaline in three redissolved solutions; (**b**) Comparison of the peak shapes of Salbutamol in three redissolved solutions.

**Figure 4 foods-12-00076-f004:**
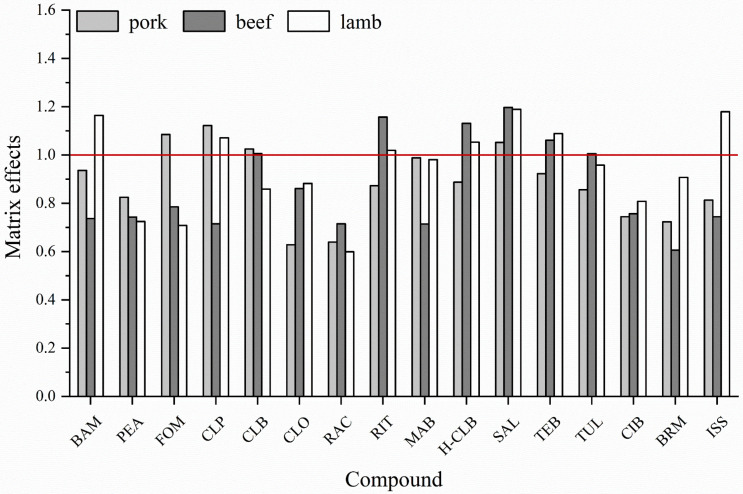
Matrix effects of sixteen *β*-agonists in three substrates. Values > 1 show matrix enhancement effect, values < 1 show matrix inhibition effect.

**Figure 5 foods-12-00076-f005:**
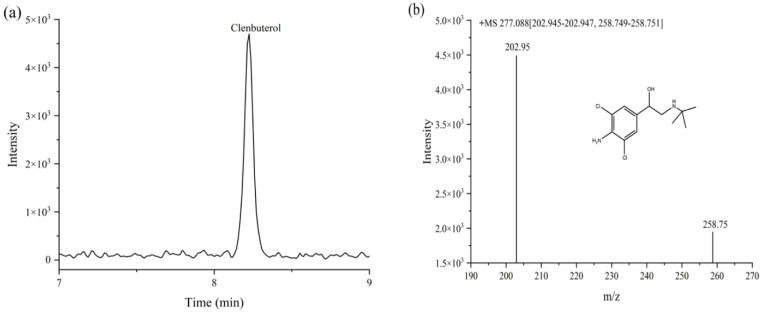
Chromatogram (**a**) and mass spectrogram (**b**) of CLB in the positive sample.

**Table 1 foods-12-00076-t001:** Mass spectrometric analysis parameters of sixteen *β*-agonists.

Compound	Internal Standard	Parent Ion (m/z)	Daughter Ion (m/z)	Collision Energy (V)	Declustering Potential (V)
CLO	CLB-D9	214.125	195.970 *	11.95	88
			154.054	17.13	
TEB	SAL-D3	226.062	152.095 *	16.04	101
			125.071	24.00	
TUL	CLB-D9	228.112	154.125 *	16.25	88
			116.429	28.80	
CIB	CLB-D9	234.162	160.125 *	14.60	92
			143.071	25.09	
SAL	SAL-D3	240.175	148.125 *	16.14	93
			222.125	10.48	
CLB	CLB-D9	277.088	202.946 *	16.04	96
			258.750	10.60	
RIT	CLB-D9	288.175	270.149 *	12.75	104
			121.071	22.35	
CLP	CLB-D9	291.088	202.958 *	15.74	97
			273.095	10.64	
H-CLB	RAC-D6	293.088	275.083 *	11.49	101
			202.958	17.81	
ISS	RAC-D6	302.175	284.196 *	13.93	100
			107.000	28.42	
RAC	RAC-D6	302.175	164.167 *	15.57	103
			284.202	11.61	
MAB	CLB-D9	311.138	237.048 *	17.01	101
			293.042	11.28	
FOM	CLB-D9	345.175	149.125 *	19.03	103
			327.125	13.38	
PEA	PEA-D3	345.225	327.036 *	12.29	102
			150.125	22.23	
BRM	CLB-D9	367.000	292.875 *	16.23	105
			348.958	11.87	
BAM	BAM-D9	368.212	294.042 *	16.82	110
			72.125	31.66	
RAC-D6		343.880	307.430	13.00	100
SAL-D3		243.162	151.054	16.00	98
CLB-D9		322.710	286.250	15.00	117
PEA-D3		347.420	330.300	12.00	102
BAM-D9		412.960	376.620	19.00	100

*: quantitative ion.

**Table 2 foods-12-00076-t002:** Effect of enzymatic condition on the recovery of sixteen *β*-agonists.

Enzymatic Condition	Number of *β*-agonists		
	Recovery < 60%	Recovery 60% to 120%	Recovery > 120%
37 °C, 12 h	0	16	0
40 °C, 2 h	0	16	0
55 °C, 2 h	7	9	0

**Table 3 foods-12-00076-t003:** Effect of extraction solvent on the recovery sixteen *β*-agonists.

Extraction Solvent	Number of *β*-agonists		
	Recovery < 60%	Recovery 60% to 120%	Recovery > 120%
No-extraction	3	10	3
Methanol	7	9	0
Acetonitrile (containing 1% acetic acid, *v*/*v*)	0	16	0

**Table 4 foods-12-00076-t004:** Effect of SPE column on the recovery of sixteen *β*-agonists.

SPE Columns	Number of β-agonists		
	Recovery < 60%	Recovery 60% to 120%	Recovery > 120%
Oasis MCX	2	14	0
Oasis HLB	3	11	2
Bond Elut C18	3	13	0
QVet-AG	0	16	0

**Table 5 foods-12-00076-t005:** Linear equations, LODs, and LOQs of sixteen *β*-agonists in livestock meat.

Matrix	Compound	Regression Equation	R^2^	Linear Range (μg/L)	LOD (μg/kg)	LOQ (μg/kg)
Pork	BAM	*y* = 0.1258*x* − 0.0057	0.9998	0.1–50	0.10	0.30
	PEA	*y* = 0.6074*x* + 0.2572	0.9928	0.1–50	0.11	0.38
	FOM	*y* = 0.1844*x* + 0.0074	0.9978	0.1–50	0.06	0.15
	CLP	*y* = 1.7030*x* − 1.0950	0.9972	0.1–50	0.03	0.08
	CLB	*y* = 1.0860*x* + 0.2131	0.9989	0.1–50	0.03	0.08
	CLO	*y* = 1.6150*x* + 0.1295	0.9965	0.1–50	0.04	0.11
	RAC	*y* = 16.220*x* − 0.9442	0.9988	0.1–50	0.08	0.24
	RIT	*y* = 0.6781*x* − 0.2005	0.9984	0.1–50	0.02	0.07
	MAB	*y* = 0.1564*x* + 0.0129	0.9988	0.1–50	0.06	0.17
	H-CLB	*y* = 13.340*x* − 5.9060	0.9979	0.1–50	0.02	0.07
	SAL	*y* = 1.4730*x* + 0.4359	0.9969	0.1–50	0.04	0.12
	TEB	*y* = 1.6870*x* + 0.2761	0.9985	0.1–50	0.08	0.23
	TUL	*y* = 2.2080*x* − 0.7053	0.9964	0.1–50	0.03	0.08
	CIB	*y* = 2.0570*x* − 1.1800	0.9968	0.1–50	0.03	0.08
	BRM	*y* = 0.0483*x* − 0.0141	0.9980	0.1–50	0.03	0.08
	ISS	*y* = 8.1590*x* − 0.4464	0.9944	0.1–50	0.01	0.04
Beef	BAM	*y* = 0.0936*x* + 0.0092	0.9994	0.1–50	0.03	0.08
	PEA	*y* = 0.4056*x* + 0.1524	0.9989	0.1–50	0.03	0.08
	FOM	*y* = 0.1063*x* + 0.0317	0.9991	0.1–50	0.03	0.08
	CLP	*y* = 0.7935*x* + 0.4343	0.9969	0.1–50	0.02	0.06
	CLB	*y* = 0.8907*x* + 0.0041	0.9994	0.1–50	0.03	0.08
	CLO	*y* = 1.4230*x* + 0.2680	0.9992	0.1–50	0.05	0.16
	RAC	*y* = 10.590*x* + 0.6483	0.9999	0.1–50	0.03	0.08
	RIT	*y* = 0.2399*x*-0.0052	0.9991	0.1–50	0.05	0.16
	MAB	*y* = 0.0834*x* + 0.0318	0.9981	0.1–50	0.04	0.13
	H-CLB	*y* = 8.1500*x* + 0.4237	0.9999	0.1–50	0.03	0.08
	SAL	*y* = 1.4270*x* + 0.0643	0.9995	0.1–50	0.02	0.06
	TEB	*y* = 1.6510*x* + 0.1169	0.9998	0.1–50	0.03	0.08
	TUL	*y* = 2.0850*x* − 0.0947	0.9991	0.1–50	0.03	0.08
	CIB	*y* = 1.4520*x* + 0.2382	0.9998	0.1–50	0.03	0.08
	BRM	*y* = 0.0312*x* + 0.0094	0.9998	0.1–50	0.03	0.08
	ISS	*y* = 6.7130*x* + 2.5610	0.9982	0.1–50	0.03	0.08
Lamb	BAM	*y* = 0.0981*x* − 0.0065	0.9997	0.1–50	0.03	0.08
	PEA	*y* = 0.3109*x* + 0.1793	0.9970	0.1–50	0.06	0.19
	FOM	*y* = 0.1031*x* + 0.0189	0.9991	0.1–50	0.03	0.08
	CLP	*y* = 1.2570*x* − 0.0721	0.9987	0.1–50	0.02	0.07
	CLB	*y* = 0.8248*x* + 0.0322	0.9995	0.1–50	0.04	0.13
	CLO	*y* = 1.5840*x* − 0.2309	0.9997	0.1–50	0.05	0.15
	RAC	*y* = 11.350*x* − 2.6370	0.9981	0.1–50	0.07	0.22
	RIT	*y* = 0.2302*x* + 0.0225	0.9998	0.1–50	0.08	0.23
	MAB	*y* = 0.1247*x* − 0.0041	0.9995	0.1–50	0.03	0.08
	H-CLB	*y* = 8.3590*x* − 2.6760	0.9971	0.1–50	0.04	0.12
	SAL	*y* = 1.4630*x* − 0.0553	0.9989	0.1–50	0.09	0.28
	TEB	*y* = 1.6790*x* + 0.1416	0.9988	0.1–50	0.03	0.08
	TUL	*y* = 2.1110*x* + 0.3056	0.9986	0.1–50	0.01	0.04
	CIB	*y* = 1.6940*x* − 0.2312	0.9997	0.1–50	0.02	0.05
	BRM	*y* = 0.0507*x* + 0.0002	0.9997	0.1–50	0.03	0.08
	ISS	*y* = 12.880*x* − 3.8780	0.9981	0.1–50	0.07	0.21

## Data Availability

All available data are contained within the article.

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
