# Peer review of "Rapid and High-Throughput Determination of Sixteen β-agonists in Livestock Meat Using One-Step Solid-Phase Extraction Coupled with UHPLC-MS/MS"

_foods, 2022, doi:10.3390/foods12010076_

Round 1

Reviewer 1 Report

This is a review of the manuscript entitled 'Rapid and High-Throughput Determination of Sixteen β-agonists in Livestock Meat Using One-Step Solid-Phase Extraction Coupled with UHPLC-MS/MS'. In this paper, authors develop a sample preparation UHPLC-MS/MS method for the determination of β-agonists in meat.

The paper is extremely well written. It was a pleasure to read. Its length accords to its importance. The analytical quality is extraordinary. The tables are needed. Images too and are of good quality and self-explanatory. References are updated (46% less than 5 years, 76% less than 10 years) but very polarized: 68% cite papers from Chinese research groups. A broader origin of citations would highlight the importance of the topic.

The main concern about the paper is its motivation. The topic is of broad interest, but probably not so novel anymore. A quick search reveals several UHPLC methods. It is not so difficult to create a faster method in UHPLC, but to get it approved by authorities for official analysis. Is the improvement in the method enough for granting this approval? For example, peaks 9 and 10 are completely overlapping even if the structure of the analytes are very different. Their molecular weight differs in 0.05 Da, and the daughter ions in 1 and 0.089. Pair 14 - 15 is partially overlapping and 6 - 7 too. My guess is that doubling the flowrate would only lead to the coelution of those peaks, which could be anyhow MS resolved, due to larger mass differences. So the 10 min shorter method compared to the Chinese standard cited are nor granting the publication.

While authors claim that their method is a one-step SPE, they do not comment in the title the need for enzymatic digestion and evaporation. How is the total time of analysis compared to other methods? A fast check returns methods that use glucuronidase at 65° 1h. That is faster (Line 368). Other use Quechers. Also fast. The Qvet-AG SPE returns only results in Chinese. Images are always showing pictures of the poly divinylbenzene -co- N-methylpirolidone. Is that its chemistry? Why did it then show differences with Oasis? If the SPE is performed in one step, I understand that the impurities will be trapped and the analytes pass thorough the column. In this case there is no enrichment (line 290), only matrix cleanup.

Summarizing, the paper is very well written, but I do not think that the method herein presented is actually faster or simpler than stablished methods. Maybe a quantitative comparison with most used methods could be helpful.

Small details:

Line 134. What does Arb mean?

L165. What do concentrations gradients mean?

L234. Peaks 9 and 10 are not separated

L286. SPE is already stablished.

L341 Recoveries between 60 and 120% is a quite broad range for recoveries. Was any factor pointed out as cause for this? The first google hit reports recoveries between 70 and 120%

L352 Table A4 is almost empty. No need a table for showing only one compound. Could be done in text.

Reviewer 2 Report

The title „ Rapid and High-Throughput Determination of Sixteen β-agonists in Livestock Meat Using One-Step Solid-Phase Extraction Coupled with UHPLC-MS/MS” wery well reflects the content of the work, which is a precise and methodically well-planned validation HPLC-MS/MS analysis and I have no comments about the work, just only 1 question:

 Line 148:  What kind of material inside SPE column was used?

Reviewer 3 Report

The article entitled "Rapid and high-throughput determination of sixteen β-agonists in livestock meat using one-step solid-phase extraction coupled with UHPLC-MS/MS" by Yang et al., is an interesting article that can really help in the detection of β-agonists in livestock meat, having successfully optimized the process. The article is very well written and justified, as well as providing adequate references. For all these reasons, I think that the article should be  the minor revisions exposed below.

Abstract

Line 21: Leave space between "Meat." and "The"?

Introduction

Line 68: Add a "," after [15].

Line 68 and line 81: "UHPLC-MS/MS" instead of "UHPLC/MS/MS", in accordance with the rest of the manuscript.

Results and Discussion

Line 218: You could provide the data of the column tested "Hypersil GOLD aQ column", in the same way as the previous two columns?

Tables

Table A3: Remove the capital letter from the word "Lamb".
